# A role for subducted albite in the water cycle and alkalinity of subduction fluids

Gil Chan Hwang [1,4], Huijeong Hwang [1,4], Yoonah Bang [1,4], Jinhyuk Choi [1], Yong Park [2], Tae-Yeol Jeon[3], Boknam Chae[3], Haemyeong Jung [2] & Yongjae Lee [1✉]

Albite is one of the major constituents in the crust. We report here that albite, when subjected to hydrous cold subduction conditions, undergoes hitherto unknown breakdown into hydrated smectite, moganite, and corundum, above 2.9 GPa and 290 °C or about 90 km depth conditions, followed by subsequent breakdown of smectite into jadeite above 4.3 GPa and 435 °C or near 135 km depth. Upon the hydration into smectite, the fluid volume of the system decreases by ~14 %, whereas it increases by ~8 % upon its dehydration into jadeite. Both the hydration and dehydration depths are correlated to increases in seismicity by 93 % and 104 %, respectively, along the South Mariana trench over the past 5 years. Moreover, the formation of smectite is accompanied by the release of $OH^-$ species, which would explain the formation of moganite and expected alkalinity of the subducting fluid. Thus, we shed new insights into the mechanism of water transport and related geochemical and geophysical activities in the contemporary global subduction system.

[1] Department of Earth System Sciences, Yonsei University, Seoul, Korea. [2] School of Earth and Environmental Sciences, Seoul National University, Seoul, Korea. [3] Beamline Science Division, Pohang Accelerator Laboratory, Pohang, Korea. [4] These authors contributed equally: Gil Chan Hwang, Huijeong Hwang, Yoonah Bang. ✉email: yongjaelee@yonsei.ac.kr

Subduction zones are important geochemical and geophysical interfaces as oceanic plates sink deep into the Earth transporting and cycling rocks and volatile components, such as water and carbon dioxide, between the surface and the mantle[1–4]. Understanding the mineral–water interaction along subduction zone conditions is therefore critical as it is related to metamorphism, magma generation, and earthquake triggering along the subducting plate[3,5–11] and in the overlying mantle wedge[2]. Alkali feldspars are important components of the subducting oceanic plate that have been used to model the generation of magma and the evolution of lithospheric mantle composition[12–14].

Amongst alkali feldspars, albite [NaAlSi$_3$O$_8$] accounts ~16% and ~10% in volume in basalt and gabbro in the oceanic crust, respectively[8]. Generally, anhydrous albite has been reported to transform to jadeite and quartz under high temperature–pressure conditions in the range of 1–3 GPa and 200–1000 °C[7,15,16]. On the other hand, Shen and Keppler reported complete miscibility between silicate melts and hydrous fluids in the near surface conditions from visual observations in the albite–H$_2$O system[13].

A series of mineral transformations have been established along diverse subduction geotherms in relation to the observed metamorphic facies[17,18]. Dehydration and breakdown of hydrous minerals down to ca. 240 km depth are believed to impact the chemical and physical properties of subducting rocks[9,19]. It is uncertain, however, how mineral transformation is related to the chemistry of subduction fluids, specially its mildly alkaline nature. Galvez et al. reported that the degree of ΔpH may be caused by intramantle differentiation or infiltration of fluids enriched in alkali components extracted from the subducted crust[3]. In this study, we report the first experimental evidence of alkaline fluid generation from the breakdown reaction in the anhydrous albite–water system under cold subducting slab conditions down to about 170 km in depth. We found that starting at 2.9 GPa and 290 °C or about 90 km depth along cold subducting slab, albite breaks down to form a mixture of smectite, moganite, and corundum, which releases OH$^-$ species in a 5:4 molar ratio. We suspect that this would impact the pH of subducting fluid by as much as ΔpH of 2.2–2.5, which is extrapolated from the calculation of ΔpH of solutions in equilibrium with subducting basaltic slab[3].

In this work, we performed in situ high-pressure and -temperature synchrotron X-ray diffraction (XRD) experiments under three different sample conditions using a resistive-heating diamond anvil cell (DAC): (1) up to 5.4 GPa and 570 °C (cold subduction slab and wet condition, down to ca. 170 km depth); (2) up to 3.2 GPa and 530 °C (warm subduction slab and wet condition, down to ca. 100 km depth); (3) up to 4.7 GPa and 505 °C (cold subduction slab and dry condition, down to ca. 145 km depth). In wet condition experiments, powdered samples of albite were immersed in water in a ca. 1:1 by volume ratio as a pressure-transmitting medium (PTM). In dry condition experiment, no PTM was added to anhydrous albite.

## Results

### In situ high-pressure and high-temperature studies using a DAC.
Upon increase in pressure and temperature in wet and cold subduction conditions following the geothermal gradients along the South Mariana trench[20], we observed that albite breaks down to hydrated Na-smectite (beidellite), moganite, and corundum above 2.9 GPa and 290 °C or about 90 km depth, as below (Fig. 1 and Supplementary Fig. 1).

$$5NaAlSi_3O_8 + 4H_2O \rightarrow NaAl_2(Si_3Al)O_{10}(OH)_2 \cdot H_2O + 12SiO_2$$
$$+ Al_2O_3 + 4Na^+ + 4OH^- \quad (1)$$

Albite 1W Na − smectite (beidellite) Moganite Corundum

The appearance of the characteristic (001) diffraction peak at $d_{001}$ ~12 Å indicates that the newly formed Na-smectite is

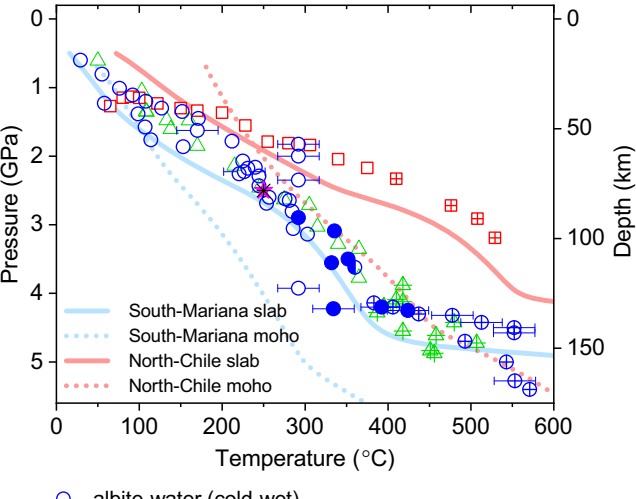

**Fig. 1 Stability of albite along three different PT conditions.** Thermal gradients of the Mariana (cold) and the North Chile (warm) slabs are shown in blue and red lines, respectively, following the W1300 model by Syracuse et al.[20]. Dotted lines are geotherms for 7 km beneath the respective slab surface. Horizontal error bars show the uncertainties in temperature.

hydrated with one layer of water with a total water content of ca. 9.0 wt.% (Fig. 2a). In a smectite clay structure, it has been established that the interlayer is mono-hydrated (1W model) when $d_{001}$ is in the range between 11.8–12.9 Å[21,22]. The observed transformation of albite along the hydrous cold subduction conditions thus reveal hitherto unknown breakdown reaction to form a hydrated clay mineral, which is quite different from the established breakdown scheme of albite to jadeite and quartz in anhydrous medium above $P = 0.035 + 0.00265\,T$ (°C) ± 0.05 GPa[7,15,16].

The other crystalline by-product along with the hydrated smectite, moganite, is known to be a stable polymorph of quartz under alkaline environment in the pH range between 9.5 and 10.5 (or ΔpH of 2.5–3.5 range)[23]. In fact, the breakdown reaction (1) requires the release of Na(OH), which is expected to be dissolved in the remaining fluid medium under our experimental conditions and hence not detectable in the observed XRD pattern. Further increase in pressure and temperature above 4.3 GPa and 435 °C, i.e., near 135 km depth conditions, results in the breakdown of the hydrated Na-smectite as the characteristic $d_{001}$ ~12 Å peak disappears[24], as below (Fig. 2).

$$NaAl_2(Si_3Al)O_{10}(OH)_2 \cdot H_2O \rightarrow NaAlSi_2O_6 + SiO_2 + Al_2O_3 + 2H_2O$$
1 W Na − smectite(beidellite) Jadeite Moganite Corundum

$$(2)$$

Interestingly, when the thermal gradient follows that of warm subduction slab in the presence of water as a PTM, we observed the partial breakdown of albite into jadeite (NaAlSi$_2$O$_6$) and α-quartz (SiO$_2$) above 2.2 GPa and 375 °C or ca. 70 km depth

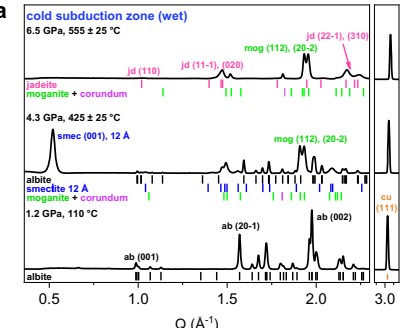
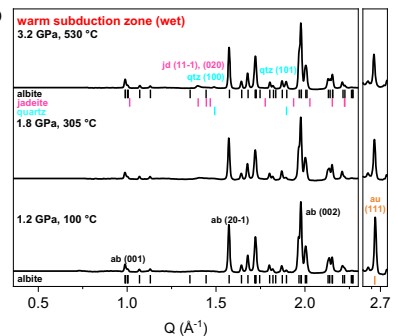
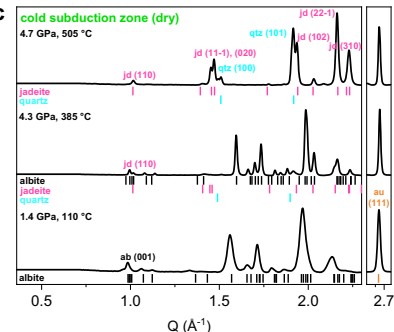

**Fig. 2 In situ X-ray diffraction patterns representing the stability of albite under three different PT conditions. a** Hydration and dehydration breakdowns from albite under aqueous cold subduction conditions up to 6.5 GPa and 555 °C. The tick marks indicate albite (black, triclinic C$\bar{1}$)[45], Na-smectite (blue, monoclinic C2/m)[46,47], jadeite (pink, monoclinic C2/c)[48], moganite (green, monoclinic I2/a)[49], and corundum (purple, rhombohedral R$\bar{3}$c)[50]. The copper and gold peaks were used as pressure scales (orange, cubic Fm3m)[51,52]. **b** Albite undergoes partial breakdown into jadeite (NaAlSi$_2$O$_6$) and α-quartz (SiO$_2$) without forming smectite under aqueous warm subduction conditions up to 3.2 GPa and 530 °C. **c** Albite undergoes previously known breakdown into jadeite and α-quartz under anhydrous cold subduction conditions up to 4.7 GPa and 505 °C. Phase abbreviations: albite: ab; smectite: smec; moganite: mog; jadeite: jd; and quartz: qtz.

conditions but could not observe the formation of the hydrated Na-smectite from albite up to 3.2 GPa and 530 °C or ca. 100 km depth conditions (Figs. 1, 2b, and Supplementary Fig. 3b). When the experiments are repeated under dry conditions along the cold subduction zone, we also observed the previously known breakdown of albite into jadeite and α-quartz above 4.3 GPa and 385 °C or ca. 135 km depth conditions (Figs. 1, 2c, and Supplementary Fig. 3c)[15,25]. As for the formation of the quartz, it has been known that the compression of the α-quartz would bypass the stability field of coesite under certain non-hydrostatic compression[26]. Overall, our experimental results manifest that the hydration breakdown of albite into smectite and release of Na (OH) is specific for the hydrous cold subduction environments (Figs. 1, 2a, and Supplementary Fig. 3a) and hence the contemporary global subduction system[20].

**Scanning electron microscopy and FT-IR measurements on the starting and recovered samples.** In order to confirm the hydration breakdown reaction of albite into the hydrated smectite and concurrent release of Na$^+$ and (OH)$^-$ into the surrounding fluid medium, scanning electron microscopy (SEM) and FT-IR measurements were performed on the starting and recovered samples. The SEM images show unambiguous transformation in the crystal morphology from albite into the smectite clay in the recoverd sample under wet conditions (Fig. 3). Mid-IR spectra from the recovered sample also reveal new features in absorbance spectrum at 3484 cm$^{-1}$ and 3587.8 cm$^{-1}$ (inset in Fig. 3b). The latter vibration is identified as the IR-active stretch mode of OH in Na(OH)[27], while the former broad streching band indicates the presence of interlayer water molecules in the 1W Na-smectite[28]. We thus confirm the reaction (1) where anhydrous albite under hydrous cold subduction conditions breaks down into a hydrated clay mineral, a quartz polymorph, and corundum by releasing Na$^+$ and (OH)$^-$ species to form alkaline fluid environment. Up to our knowledge, this is the first experimental observation of breakdown into a hydrated clay and release of alkaline species from subducting oceanic crustal mineral[29]. Considering the estimated content of albite in basalt, i.e., ~16% in volume, in the subducting oceanic crust, our experimental design of the 1:1 volume mixture of albite and water translates to the water contents of ca. 5 wt.% H$_2$O, which is assumed to be the water content in the cold subduction zone[9,30,31]. Furthermore, the formation of

the hydrated Na-smectite from albite is confirmed by XRD and SEM measurements on the quenched samples from ex situ large volume press syntheses at 2.4 GPa and 260 °C conditions using lower water-to-sample ratios from 50 wt.% H$_2$O down to 6 wt.% H$_2$O (Fig. 3c and Supplementary Data Fig. 2). Our lowest water-to-albite ratio of 6 wt.% H$_2$O would then translate to the basalt with ca. 0.9 wt.% H$_2$O. According to van Keken et al.[9], the water content of basalt at 90 km depth in a cold subducting slab, which is the depth of our hydration breakdown of albite, would be maintained to be over 2 wt.%, while warm subducting slabs would be almost dehydrated by ca. 90 km. Therefore, our experimental conditions would simulate that if hydrous minerals contribute water down to ca. 90 km depth to make the cold subducting slab wet, the hydration breakdown of albite into smectite would occur.

**Ex situ high-pressure and high-temperature studies with a modified Griggs apparatus.** In order to complement our experimental findings using a single mineral phase in a DAC, we have expanded our investigation into a macroscopic scale using a natural basalt rock containing ca. 66 vol.% of feldspars. Two sets of 3 mm diameter core-drilled samples of a basalt were heated at 250 °C and 2.5 GPa for 10 h using a modified Griggs apparatus (Fig. 1). In order to simulate hydrous conditions expected along the cold subduction system, one set was contained with ca. 5.5 wt. % of water in a sealed platinum capsule, while the other was treated without water. The recovered sample from the wet run showed that sample has been severely eroded, compared to the fresh sample from the dry run (Fig. 3). XRD data clearly revealed the presence of ~12.9 Å peak in the sample recovered from the wet run (Supplementary Fig. 4).

## Discussion

The observed sequential breakdowns from albite lead to significant changes in the net crystalline density (see "Methods" and Supplementary Table 1). When albite breaks down into the smectite mixture at ca. 90 km depth conditions, the net crystalline density increases, i.e., from 2.58(1) g cm$^{-3}$ to 2.63(1) g cm$^{-3}$, as the 1W Na-smectite, moganite, and corundum form with densities of 1.76 (1) g cm$^{-3}$, 2.62(1) g cm$^{-3}$, and 3.90(1) g cm$^{-3}$, respectively, in the molar ratio shown in the reaction scheme (1). Subsequently upon the breakdown of the 1W Na-smectite at ca. 135 km depth conditions, the net crystalline density then increases by ca. 11%

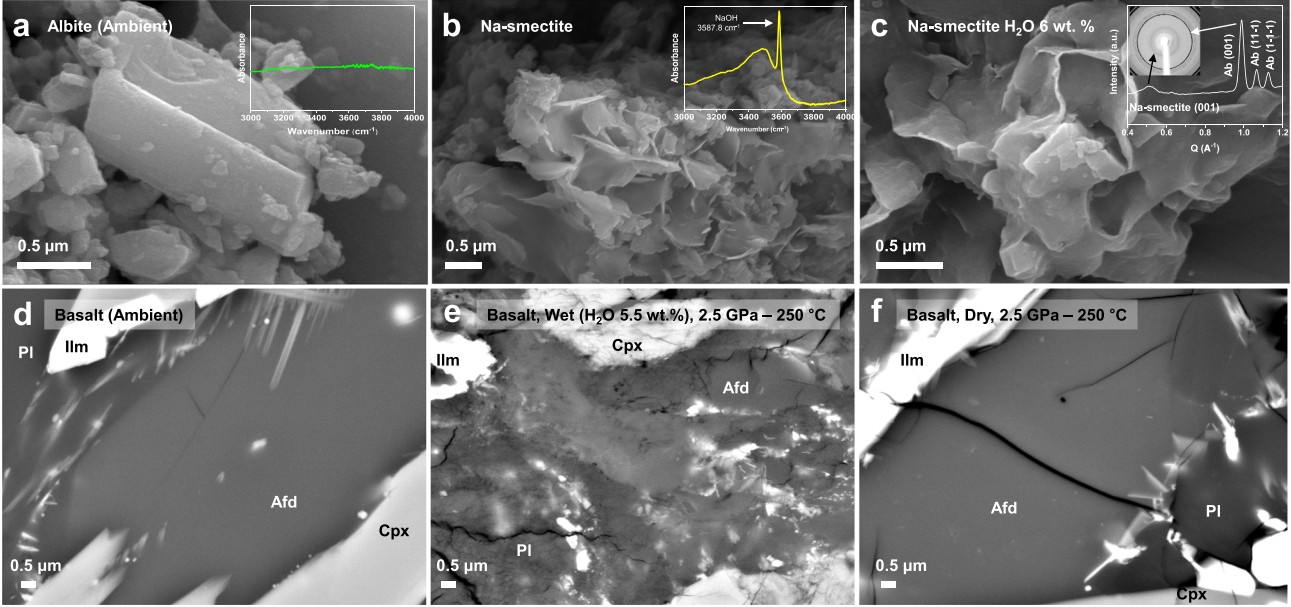

**Fig. 3 SEM images of the original albite (basalt) and its recovered products after different PT treatments.** Mid-IR spectra from respective samples are shown as inset images. **a** The original anhydrous albite is characterized by its prismatic crystal habits and featureless IR spectra (upper inset) whereas **b** the recovered product under 1:1 albite-to-water ratio shows the characteristic clay crystal habits and the IR band (upper inset) by NaOH at 3587.8 cm$^{-1}$ (ref. [27]). The O–H stretching mode of the hydrated Na-smectite is also identified as a broad band at 3484 cm$^{-1}$. **c** A representative SEM image and XRD pattern (upper inset) of the recovered product using 6 wt.% H$_2$O from 2.4 GPa and 260 °C conditions (see Supplementary Fig. 2 for data using different water contents). **d** A representative SEM image of the original natural basalt (starting material). **e** A representative SEM image of the recovered basalt sample from 2.5 GPa and 250 °C under wet condition by 5.5 wt% H$_2$O. **f** A representative SEM image of the recovered basalt sample from 2.5 GPa and 250 °C under dry condition. Phase abbreviations: alkali-feldspar: Afd; plagioclase: Pl; clinopyroxene: Cpx, augite; and ilmenite: Ilm.

from 2.68(1) g cm$^{-3}$ to 2.98(1) g cm$^{-3}$. On the other hand, if we assume the starting volume of water is the same as that of albite, the net fluid volume would decrease by ca. 14%, e.g., from 1 m$^3$ to 0.86 m$^3$, upon the formation of the 1W Na-smectite at ca. 90 km depth, and then increase by ca. 8%, e.g., from 0.86 m$^3$ to 0.93 m$^3$, upon the breakdown of the 1W Na-smectite at ca. 135 km depth (Supplementary Table 1). The increase in the net fluid volume would correspond to ca. 9 wt.% release of water, which is higher than the amount of water released by the dehydration breakdown of hydrated subducting minerals such as phengite and amphibole, i.e., ca. 4.3 wt.% and 2 wt.%, respectively[32]. When we extend the estimated changes of the net density and fluid volume to subducting basalt composed of ~16 vol.% of albite, the degree of changes in the net density is modulated compared to those based on albite alone, i.e., the net density of the basalt is 3.16(1) g cm$^{-3}$ at ambient condition, increases to 3.17(1) g cm$^{-3}$ upon the formation of the hydrated smectite at ca. 90 km depth, and then further increases to 3.22 g cm$^{-3}$ upon the breakdown of the hydrated smectite into jadeite and quartz at ca. 135 km depth. However, the degree of changes in the net fluid volume is comparable to the changes estimated based on albite alone, i.e., concomitant to the discontinuous changes in the net density of basalt, the net fluid volume decreases by ca. 11.5% and then increases by ca. 6%. Such concerted and discontinuous changes in the net fluid volume and crystalline density via hydration (reaction 1) and subsequent dehydration (reaction 2) breakdowns would impact the physical properties of the subducting slab itself and hence the earthquake-triggering mechanism in the depth range between ca. 90 and 135 km along the cold subducting slabs.

Seismic data along the South Mariana trench are in line with our observations that the intermediate-depth earthquakes increase by ca. 93% as the depth range increases from 70–90 km to 90–110 km (International Seismological Centre, ISC Bulletin),

which is close to the region of the observed hydration breakdown of albite into a smectite clay. Ikari and Kopf, and Hirono et al. have recently shown that the presence of low-friction clay minerals such as smectite may play an important role in the dynamic weakening processes[33,34]. Based on the friction experiments simulating plate convergence rates of centimeters per year, they discovered unstable slip behavior in clay-dominated faults. The low shear strength of weak clay minerals in clayey fault would favor earthquake rupture accompanying large slip. Our results thus extend such a clay-mediated earthquake generation to deeper depths along subduction zone. At the depth range of 110–130 km, the frequency of earthquake occurrence decreases in comparison to the 90–110 km region. In the 130–150 km region, the earthquake frequency increases again by ca. 104%, which is the region where the dehydration breakdown of the smectite clay into jadeite occurs (Fig. 4). This can be well associated with the established dehydration embrittlement theory for intermediate-depth earthquakes[35]. In order to distinguish our new finding of the formation of hydrated clay mineral along cold subduction zone and its association with seismicity increase from dehydration embrittlement, we propose a new term, "hydration faulting (or smectization faulting)" as a possible new mechanism of subduction zone earthquakes. We encourage future studies to test our hypothesized mechanism that the increased seismicity is mainly from this hydration-related dynamic weakening process, and such phenomenon can be found in other subduction zones.

Furthermore, our results help us gain new insights into the geochemical processes occurring along cold subduction zones where fluid is transported deep into the mantle (Fig. 4)[9,36]. The extrapolated ΔpH of fluid in equilibrium with subducting basaltic slab[3] to the temperature where the hydration breakdown of albite into a smectite clay has been observed is ca. 2.5, which is within the ΔpH range for the concurrent formation of moganite. The

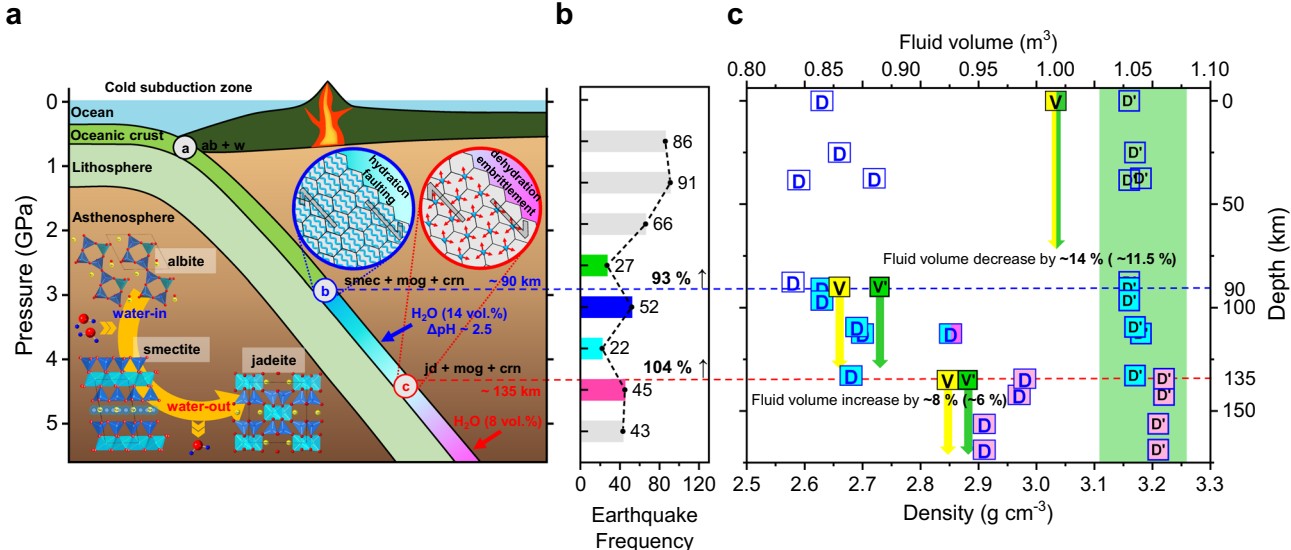

**Fig. 4 The breakdown model of albite along the South Mariana trench. a** The hydration and dehydration breakdowns of albite occur at ca. 90 km and 135 km depths, respectively. **b** The frequency of earthquake occurrence along the South Mariana trench over past 5 years (ISC Bulletin, investigation period between Nov. 2014 and Nov. 2019). **c** Changes in the net crystalline density (D for albite, D' for basalt) and fluid volume (V for albite, V' for basalt) in the subducting system are estimated as a function of subducting depth as in **a** and **b**. Green arrow and region highlight the estimated changes based on a basalt system. Phase abbreviations: albite: ab; smectite: smec; moganite: mog; corundum: crn; jadeite: jd; and water: w.

origin of calc-alkaline magma in the overriding mantle wedge might also be related to the breakdown of albite and release of Na (OH) to the surrounding fluid[37].

Our results also shed new insights into interpreting anomalies in electrical and seismic properties along cold subduction zones. It has been suspected that the presence of hydrous minerals and/or increased melt/fluid fractions enhance electrical conductivity and seismic wave attenuation along with the thermal gradient of the subduction zone[38]. The conductivity model by Ni et al. and Pommier et al. proposes that the addition of ca. 7 wt.% $H_2O$ to basalt at 1200 °C explains the difference in electrical conductivity between North Chile (ca. 0.04–0.1 S m$^{-1}$) and Mariana (ca. 0.01 S m$^{-1}$)[38,39]. It has been known that warm slab, such as North Chile slab, may have released most of its aqueous fluids by ca. 90 km depth, whereas cold slab, such as Mariana slab, can still hold the fluids beyond 90 km depth, enriching partial melt accumulation zones with aqueous phases and leading to lower conductivity values[39]. Our observation on albite subjected to such hydrous cold subduction conditions supports these by showing the hydration breakdown of albite into smectite clay near ca. 90 km depths removing ca. 14 vol.% of fluid from the region (assuming a 1:1 mixture of albite and water) and then subsequent dehydration breakdown of smectite into jadeite above ca. 135 km depth releasing ca. 8 vol.% of fluid into the region.

## Methods
**Sample**. The albite sample used in our DAC experiments is from Cheshire county in New Hampshire, USA. Initial elemental characterization was performed using X-ray fluorescence (XRF, M4 Tornado, Bruker Co.) and XRD (MicroMax-007HF, Rigaku Corp.). For XRF, a combination of 25 μm diameter X-ray from poly-capillary lens and XFlash silicon drift detector was used to measure relative elemental concentrations at several points on a bulky albite crystal in vacuum. The composition of the albite sample was derived to be $Na_{1.05}K_{0.002}Ca_{0.001}Al_1Si_{2.98}O_8$. XRD was performed using a combination of 200 μm diameter X-ray (Mo-K$\alpha_1$, $\lambda$ = 0.709317(1) Å) and an imaging plate (IP) detector (R-axis IV$^{++}$, Rigaku Corp.) in transmission geometry. The sample-to-detector distance was 150 mm and the exposure time was 10 min. Sample was contained in a boron capillary with an inner diameter of 0.3 mm and rotated between 0 to 180° during measurement for better powder averaging. LaB$_6$ (NIST SRM 660c, $a$ = 4.15704(8) Å) standard powder was used to derive the instrumental parameters. IPanalyzer v3.8 program was used to calibrate and process the 2D diffraction images.

**Scanning electron microscopy**. Field emission scanning electron microscopy (FE-SEM, JEOL-7800F) was performed on the original albite (ca. ×40,000) and its recovered products (ca. × 20,000). Each sample was coated in platinum (Sputter coater 108auto, Cressington Scientific Instruments) and measured for 120 s in vacuum. The images were measured under LED (lower electron detector) mode at 10 kV.

**High-pressure and high-temperature synchrotron XRD**. In situ high-pressure and -temperature XRD experiments were performed at the 3D beamline at Pohang Accelerator Laboratory (PAL). Monochromatic X-ray with 0.6888(1) Å (18 keV) in wavelength and 100 μm in size was provided using a DCM (double crystal monochromator) of bent Si (111) and Si (311) crystals. Angle-dispersive XRD data were measured using an image plate detector (MarXperts Mar345, 3000 × 3000 pixels, exposure time of 400 s). CeO$_2$ (NIST SRM 674b, $a$ = 5.4115(3) Å) standard powder was used for the instrument calibration. As a pressure vessel, a symmetric-type diamond-anvil cell (SDAC) was used in combination with a membrane-driven pressure control system (GE Pace5000). The culet diameter of the diamond anvil (type I) was 500 μm. A rhenium gasket was used after indentation to about 100 μm in thickness. A sample chamber of 150 μm in diameter was made in the gasket using an electric discharge machine (EDM; Holozoic products). Pure water was used as a PTM in the case of aqueous conditions experiments. Simultaneous P-T condition was created by using a set of house-made resistive coil heater (Kanthal-A1 wire of 26 gauge and $\phi$ 0.4 mm with a DC power supply by Sorensen XG 60-14, Ametek Inc.) to surround each diamond anvil[40]. Temperature was monitored using a K-type thermocouple attached to the pavilion of the diamond anvil close to the sample with the maximum uncertainties of ±3 °C[41], and the sample pressure was estimated using the equation of state of the Au pressure marker. XRD at each PT step was measured for 400 s while rocking the DAC by 4°.

**Infrared spectroscopy**. Synchrotron IR experiments were performed at the beamline 12D at PAL. FT-IR spectra were measured in the mid-IR range (700–7000 cm$^{-1}$) using a Vertex 80v + Hyperion 3000 microscope with ×15 objective and a mercury cadmium telluride (MCT) detector (Bruker Co.). Infrared spectra obtained using synchrotron IR beam with an aperture of 50 × 50 μm$^2$, and 128 scans were measured in transmission mode while treating the background from the diamond signal using OPUS 7.5 software (Bruker Co.).

**Large volume press experiments**. Ex situ high-pressure and -temperature syntheses were performed using a QUICKpress apparatus (Depths of the Earth Co.) on a series of samples at various albite-to-water ratios from 50 wt.% down to 6 wt.% $H_2O$ using an analytical balance (Precisa XB 220A). Each albite and water mixture was loaded into a 3.4-mm diameter and 12-mm height Teflon capsule. A piston of 10 mm diameter and a cylindrical graphite heater were used to treat the sample at 2.4 GPa and 260 °C for 1 h. Each recovered product was identified using FE-SEM and XRD as described above (Fig. 3c and Supplementary Fig. 2).

**A modified Griggs apparatus experiment and SEM-EDS measurement**. The starting material was a natural basalt rock collected from Jeongok in Korea. The sample is massive and fine-grained containing plagioclase (~56 vol.%), alkali-feldspar (~10 vol.%), clinopyroxene (augite) (~10 vol.%), olivine (forsterite) (~23 vol.%), and ilmenite (~1 vol.%). The composition of feldspars was measured using the JEOL JXA-8530F field emission electron probe microanalyzer (FE-EPMA) at the National Center for Inter-university Research Facilities at Seoul National University (SNU), Korea. The measurement conditions included accelerating voltage of 15 kV, current of 10 nA, and beam size of $10 \times 10 \ \mu m^2$. The derived chemical formula of feldspars was $K_{0.02}Na_{0.36}Ca_{0.62}Al_{1.60}Si_{2.38}O_8$ for plagioclase (labradorite), and $K_{0.32}Na_{0.58}Ca_{0.04}Al_{0.98}Si_{2.88}O_8$ for alkali-feldspar (sanidine). Using a modified Griggs apparatus[42] housed at the Tectonophysics Laboratory at the School of Earth and Environmental Sciences (SEES) at SNU, a core-drilled basalt sample with 3 mm diameter was pressurized up to 2.5 GPa over 10 h and then heated up to 250 °C in 15 min, which was held for 10 h. The experiments were performed at dry and wet (5.5 wt.% $H_2O$) conditions. A graphite heater was used, and temperature was monitored using a B-type thermocouple to the maximum uncertainties of ±10 °C. After the P-T run, the sample was quenched to room temperature. Field emission-scanning electron microscopy with energy dispersive spectroscopy (FE-SEM-EDS) measurement was carried out on the recovered core samples using the FE-SEM (JSM 7100F) and the EDS system (Oxford Instruments) at the SEES at SNU. Back-scattered electron images of the recovered samples were also obtained using the same FE-SEM (JSM 7100F), which was operated at an accelerating voltage of 15 kV and a working distance of 10 mm.

**Calculation procedures of the "net crystalline density" and "net fluid volume"**
*Net crystalline density*. Density of each phase was calculated using $\rho_{calc}$ (g cm$^{-3}$) = $(M Z V^{-1}) \times$ (Avogadro's number)$^{-1}$. $V$ (Å$^3$) is derived from profile fitting using the GSAS program[43,44]. $M$ is the molecular weight and $Z$ is the number of the formula unit per unit cell. Net crystalline density accounts the proportions of the composing crystalline phases. Net crystalline density is described here as:

$$\rho_{\text{Net crystalline density}} = a\rho_a + b\rho_b + c\rho_c + \cdots + z\rho_z$$

where $a$, $b$, $c$, …, and $z$ means the proportions of the composing crystalline phases. The calculated density of each phase, $\rho_a$, is derived from XRD data.

*Net fluid volume*. To determine the changes in the net fluid volume after the reaction, the initial volume of albite and water per mole is defined as:

$$\frac{\text{Avogadro's number}\left(\frac{\text{atom}}{\text{mol}}\right) \times \text{unit cell volume of albite}(\text{cm}^3)}{\text{The number of the formula unit per unit cell (atom)}}$$

Based on the reaction schemes of the hydration (1) and dehydration (2) breakdowns, changes in the net fluid volume are then calculated.

## Data availability
All data generated or analyzed during this study are included with this published article and its Supplementary Information.

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

## Acknowledgements

This work was supported by the Leader Researcher program (NRF-2018R1A3B1052042) of the Korean Ministry of Science, ICT and Planning (MSIP). We also thank the support of the NRF-2016K1A4A3914691, NRF-2019K1A3A7A09033395, and NRF-2020R1A2C2003765. Experiments using synchrotron radiation were supported by Pohang Accelerator Laboratory (PAL).

## Author contributions

G.H., H.H., and Y.B. contributed equally to the experiments and data analysis with the help from J.C., T.J., and B.C. Y.P. and H.J. contributed to the modified Griggs apparatus experiment. Y.L. designed the research and worked on the manuscript with all authors.

## Competing interests

The authors declare no competing interests.
