## [Peer Review File · Nature Communications]

REVIEWER COMMENTS

Reviewer #1 (Remarks to the Author):

Review to the manuscript titled "A role for subducted albite in the water cycle and alkalinity of subduction fluids"

The work documented by Hwang et al. summarized an effort to measure the breakdown of albite and its subsequent products at various physical conditions. The major observations made from a set of high-pressure high-temperature synchrotron X-ray diffraction experiments to dry and wet albite under cold/warm subduction environments. Results show clear evidence of the breakdown of albite and smectite for the cold subduction zone condition at depths of ~ 90 km and at ~ 120 - 130 km, respectively. The X-ray diffraction patterns as well as the SEM images presented in the manuscript are convincing that corresponding breakdowns have happened during the discussed experiments. Overall the technical part of the manuscript can be regarded as sound as a mineral physics experiment can go. In terms of the novelty, the experiments provide the first evidence of the hydration of the feldspar and dehydration for the smectite in the subducted crust. As an important constituent, their breakdowns at intermediate depths may contribute significantly to the (two) onsets of seismicity increase/decrease due to the hydration/dehydration (which I feel a bit confusing and skeptic) observed at the S. Mariana subduction zone. The nature of the work is of substantial importance for the interpretation of the seismicity in cold subduction zones, and perhaps will provide a pathway to understand the origin of moganite in which alkaline fluid is needed. As a result, I think the result reported here shows significance for both geochemical and geophysical importances. However, I feel that to support some of the conclusions made by the authors, clarifications and/or further calculations are needed at least from mainly two aspects and they are listed below for the authors to consider:

The experiments are designed with the 1:1 mixture of albite and water (for wet condition experiments), and the results are interpreted based on this design. However, the water brought into the subduction may be held within the fractured zones of the slab and I wonder how much of the crust is under that physical condition ($< 10\%$? Or $> 50\%$ as the water may come from the serpentine?). As a result, an argument is needed here to support the usage of the experiment result for the derivation of changes in density/fluid volume "in the subduction system". It is also difficult to apply this conclusion to completely explain the electrical conductivity differences between different subduction zones.

Mechanism in intermediate depth seismicity. One of the major motivations and significance in this study is to explain the changes in intermediate depth seismicity for the cold slab. As shown in Fig. 4, both the fluid volume decrease (hydration of albite) and increase (dehydration of smectite) are related to the increase of seismicity (93% and 104% respectively) for this subduction zone. If the seismicity change is due to the dehydration embrittlement (e.g., concluded by Hacker et al.), one would expect a pattern of decrease and increase in seismicity at the corresponding depths (the hydration of albite decreases the embrittlement at depths of ~ 90 - 110 km). The authors are advised to provide more detailed discussion and better reasoning when connecting the experiment results to the earthquake frequency (e.g., what the mechanism used to relate the weakening and water).

In addition to these major points, I'd like to see some discussion about the density variations caused by the hydration/dehydration processes observed here. How does that change the bulk density structures of the subducted slab and its dynamics may also deserve some discussion (shown in Fig. 4).

Overall, the manuscript is a fun read for a non-mineral physicist and I think it may hold potential importance to a broad spectrum of fields of the earth sciences. But some of the major conclusions

need clarifications before it is considered by Nature Communications.

Reviewer #2 (Remarks to the Author):

This paper presents a new observation on the formation of hydrated clay smectite, through the reaction between albite and H₂O at the conditions along cold subducting slab. The manuscript is clear and very well written. These results would be important for understanding the mechanism of water transport and related geochemical and geophysical processes occurring in the global subduction system. I am listing the following minor questions that might help to strengthen their conclusion. I recommend that this manuscript is published once those questions are answered to satisfactory.

Line 89

It would be better to describe the calculation procedures of "net crystalline density" and "net fluid volume" in the manuscript.

Line 312

The temperature and pressure conditions should be very important for the implication of this study. To show the reliability of the temperature measurements, I suggest to describe where the thermocouples were located in the experimental system. Also, I do not understand where Au pressure marker was located. I can not see the peaks of Au from XRD patterns (figure 2) and recovered products (figure 3).

Figure 2

Is it possible to explain the reason why the peaks of SiO₂ quartz appeared in figure 2c? It seems that coesite is thermodynamically stable SiO₂ polymorph at this condition.

Extended Data Fig. 3

Some new peaks appeared around 1.4-1.5 Å at temperatures higher than 300°C in Extended Data Fig. 3b. Also, the peaks around 1.5 Å in Fig.3a and 3c were not indexed. I suggest to explain them carefully.

The **three key issues** the reviewers raised have been addressed as below:

1. **Net crystalline density/Net fluid volume:** We described the calculation procedures of the “net crystalline density” and “net fluid volume” in the method section of the revised manuscript (p15-16, line 308-323). We also added estimated changes of net density and net fluid volume of subducting basalt composed of feldspar, olivine, pyroxene, and magnetite as key minerals. As stated in the manuscript, albite accounts ~16 % in volume in basalt, and our new estimation shows the effect of the hydration and dehydration breakdowns from albite on the subducting basaltic slab. The degree of changes in the net density of basalt is modulated compared to those based on albite alone, i.e., the net density of the basalt is 3.16(1) g/cm³ at ambient condition, increases to 3.17(1) g/cm³ upon the formation of the hydrated smectite at ca. 90 km depth, and then further increases to 3.22 g/cm³ upon the breakdown of the hydrated smectite into jadeite and quartz at ca. 135 km depth. However, the degree of changes in the net fluid volume is comparable to the changes estimated based on albite alone, i.e., concomitant to the changes in the net density of basalt, the net fluid volume decreases by ca. 11.5 % and then increases by ca. 6 %, which confirms the correlation of the hydration and dehydration breakdowns with the increases in the seismic frequencies at ca. 90 and 135 km depths, respectively (revised Fig. 4).

2. **Relationship between seismicity and hydration/dehydration:** In this revision, we have added the discussion to link our observed hydration and dehydration breakdowns occurring along cold (wet) subduction zone to the increases in the seismic frequency. The hydration breakdown of albite occurs when it transforms into the hydrated smectite mixture at ca. 90 km depth along cold (wet) subduction zone. Ikari et al. (2017) and Hirono et al. (2019) have recently shown that the presence of low-friction clay minerals such as smectite may play an important role in the dynamic weakening processes. Based on the friction experiments simulating plate convergence rates of centimeters per year, they discovered unstable slip behavior in clay-dominated fault (new references #33-34). The low shear strength of weak clay minerals in clayey fault would favor earthquake rupture accompanying large slip. Our results thus extend such a clay-mediated earthquake generation to deeper depths along subduction zone. Subsequent dehydration breakdown of the hydrated smectite into jadeite mixture at ca. 135 km depth can be well associated with the established dehydration embrittlement theory for intermediate earthquakes (new reference #35). In order to distinguish our new finding of the formation of hydrated clay mineral

along cold (wet) subduction zone and its association with seismic increase from dehydration embrittlement, we have introduced a new term, “Hydration faulting (or Smectization faulting)” as a possible new mechanism of subduction zone earthquakes (revised Fig. 4).

3. Presentation of X-ray diffraction patterns: In this revision, we show the peaks of the Au/Cu pressure markers in the XRD patterns (revised Fig. 2 and Supplementary Fig. 3). We have also added related references on the pressure scale in the figure caption (Anderson et al (1989) and Wang et al (2009), new references #46-47). As carefully pointed out by reviewer #2, we indexed the weak peaks around 1.4-1.5 (\AA^{-1}), which were identified as jadeite-moganite and jadeite-quartz in Supplementary Fig.3a and Fig.3b&c, respectively. Accordingly, we have revised the stability of albite along the warm subduction condition to be similar to that of cold (dry) subduction condition and hence the previously known breakdown into jadeite and quartz (p4, line 92, revised Fig. 1 and 2).

Full details of the reviewers’ comments and our point-by-point responses are summarized below (all the changes made in the revised version are marked in red):

Reviewer #1 (Remarks to the Author)

Comments: Review to the manuscript titled “A role for subducted albite in the water cycle and alkalinity of subduction fluids”

The work documented by Hwang et al. summarized an effort to measure the breakdown of albite and its subsequent products at various physical conditions. The major observations made from a set of high-pressure high-temperature synchrotron X-ray diffraction experiments to dry and wet albite under cold/warm subduction environments. Results show clear evidence of the breakdown of albite and smectite for the cold subduction zone condition at depths of ~ 90 km and at ~ 120-130 km, respectively. The X-ray diffraction patterns as well as the SEM images presented in the manuscript are convincing that corresponding breakdowns have happened during the discussed experiments. Overall the technical part of the manuscript can be regarded as sound as a mineral physics experiment can go. In terms of the novelty, the experiments provide the first evidence of the hydration of

the feldspar and dehydration for the smectite in the subducted crust. As an important constituent, their breakdowns at intermediate depths may contribute significantly to the (two) onsets of seismicity increase/decrease due to the hydration/dehydration (which I feel a bit confusing and skeptic) observed at the S. Mariana subduction zone. The nature of the work is of substantial importance for the interpretation of the seismicity in cold subduction zones, and perhaps will provide a pathway to understand the origin of moganite in which alkaline fluid is needed. As a result, I think the result reported here shows significance for both geochemical and geophysical importances. However, I feel that to support some of the conclusions made by the authors, clarifications and/or further calculations are needed at least from mainly two aspects and they are listed below for the authors to consider:

Comments: *The experiments are designed with the 1:1 mixture of albite and water (for wet condition experiments), and the results are interpreted based on this design. However, the water brought into the subduction may be held within the faltered zones of the slab and I wonder how much of the crust is under that physical condition (< 10%? Or > 50% as the water may come from the serpentine?). As a result, an argument is needed here to support the usage of the experiment result for the derivation of changes in density/fluid volume “in the subduction system”. It is also difficult to apply this conclusion to completely explain the electrical conductivity differences between different subduction zones.*

Reply: Considering the estimated content of albite in basalt, i.e., ~16 % in volume, in the subducting oceanic crust, our experimental design of the 1:1 volume mixture of albite and water mimics the estimated water contents of ca. 5 wt. % H₂O in a wet subducting slab. Besides, we observed the formation of the hydrated smectite under low water-to-albite ratios from 50 wt. % H₂O down to 6 wt. % H₂O (Fig. 3c and Supplementary Fig. 2). Our lowest water-to-albite ratio of 6 wt. % H₂O would then translate to the basalt with ca. 0.9 wt. % H₂O. According to van Keken et al., (2011), the water content of basalt at ca. 90 km depth in a cold (wet) subducting slab, which is the depth of our hydration breakdown of albite, would be maintained to be over 2 wt. %, while warm subducting slabs would be almost dehydrated by ca. 90 km. Therefore, our experimental conditions would simulate that if hydrous minerals contribute water down to ca. 90 km depth to make the cold subducting slab wet, the hydration breakdown of albite into smectite would occur. As pointed out in our key revision point #1, we have added the estimated changes of the net fluid volume upon the hydration and dehydration breakdowns of albite in terms of a subducting basaltic rock containing ~16 vol. % of albite in 1:1 volume ratio with water, which is comparable to the changes estimated based on albite alone and shows correlation with the changes in the seismic frequency at respective breakdown depths (p5, line 117).

Regarding the complete application of our results into the electrical conductivity differences between different subduction conditions, we are currently limited to suggest the possible impact of the changes in the net fluid/hydrous minerals contents on the electrical conductivity (p8, line 193) to encourage follow up experimental and computational works.

Comments: *Mechanism in intermediate depth seismicity. One of the major motivations and significance in this study is to explain the changes in intermediate depth seismicity for the cold slab. As shown in Fig. 4, both the fluid volume decrease (hydration of albite) and increase (dehydration of smectite) are related to the increase of seismicity (93% and 104% respectively) for this subduction zone. If the seismicity change is due to the dehydration embrittlement (e.g., concluded by Hacker et al.), one would expect a pattern of*

decrease and increase in seismicity at the corresponding depths (the hydration of albite decreases the embrittlement at depths of ~ 90-110 km). The authors are advised to provide more detailed discussion and better reasoning when connecting the experiment results to the earthquake frequency (e.g., what the mechanism used to relate the weakening and water).

Reply: This issue has been addressed in the key revision point #2 above. Furthermore, we have introduced a new term, “hydration faulting (or smectization faulting)” in this revision as a possible new mechanism of subduction zone earthquakes. As described, this is associated with the formation of hydrated smectite at extended depths along cold (wet) subduction zone (new references #33-34) and thus distinguished from the established “dehydration embrittlement” to explain intermediate-depth earthquakes (revised Fig. 4).

Comments: In addition to these major points, I'd like to see some discussion about the density variations caused by the hydration/dehydration processes observed here. How does that change the bulk density structures of the subducted slab and its dynamics may also deserve some discussion (shown in Fig. 4).

Reply: We have taken the issue on the density and fluid volume as one of our key revision points. As addressed in the key revision point #1 above, we have added the definition and calculation procedures of the net crystalline density and net fluid volume (p15-16, line 308-323) and applied them for both a single albite and a basalt system (p6, line 144, revised Fig. 4).

Comments: Overall, the manuscript is a fun read for a non-mineral physicist and I think it may hold potential importance to a broad spectrum of fields of the earth sciences. But some of the major conclusions need clarifications before it is considered by Nature Communications.

Reply: We appreciate such an encouraging remark and insightful comments by reviewer #1. We hope we have clarified all the issues raised by reviewer #1 in this revision.

Reviewer #2 (Remarks to the Author)

Comments: This paper presents a new observation on the formation of hydrated clay smectite, through the reaction between albite and H₂O at the conditions along cold subducting slab. The manuscript is clear and very well written. These results would be important for understanding the mechanism of water transport and related geochemical and geophysical processes occurring in the global subduction system. I am listing the following minor questions that might help to strengthen their conclusion. I recommend that this manuscript is published once those questions are answered to satisfactory.

Reply: We appreciate such supportive comments by reviewer #2. Our point-by-point responses to the questions are as below.

Comments: Line 89, It would be better to describe the calculation procedures of “net crystalline density” and “net fluid volume” in the manuscript.

Reply: As addressed in our key revision point #1, we have added the definition and calculation procedures of the net crystalline density and net fluid volume (p15-16, line 308-323) and applied them for both a single albite and a basalt system (p6, line 144, revised Fig. 4).

Comments: Line 312, The temperature and pressure conditions should be very important for the implication of this study. To show the reliability of the temperature measurements, I suggest to describe where the thermocouples were located in the experimental system. Also, I do not understand where Au pressure marker was located. I can not see the peaks of Au from XRD patterns (figure 2) and recovered products (figure 3).

Reply: As suggested, we added a description on the location of thermocouples in our diamond anvil cell setup with related references on the temperature calibration (p14, line 266, Mendez et al. (2020) as new reference #49). In addition, we have revised all the figures showing our DAC-XRD patterns to include the peaks of the Au/Cu pressure markers (revised Fig. 2 and Supplementary Fig. 3) with related references on the pressure scale in the caption (Anderson et al (1989) and Wang et al (2009) as reference #46-47).

Comments: Figure 2, Is it possible to explain the reason why the peaks of SiO₂ quartz appeared in figure 2c? It seems that coesite is thermodynamically stable SiO₂ polymorph at this condition.

Reply: We added a reference to explain the appearance of quartz in the stability region of coesite. In Carl et al. (2017), the compression of the α -quartz would bypass the stability field of coesite under certain non-hydrostatic compression conditions.

Comments: Extended Data Fig. 3, Some new peaks appeared around 1.4-1.5 Å at temperatures higher than 300°C in Extended Data Fig. 3b. Also, the peaks around 1.5 Å in Fig.3a and 3c were not indexed. I suggest to explain them carefully.

Reply: We have taken this comment as our key revision point #3. As explained above, we indexed the weak peaks around 1.4-1.5 (Å⁻¹) in Supplementary Fig. 3 to be jadeite-moganite (S. Fig. 3a) and jadeite-quartz (S. Fig.3b and 3c). This led us revise the stability of albite along the warm subduction condition to be similar to that of cold (dry) subduction condition and hence the previously known breakdown into jadeite and quartz (p4, line 92, revised Fig. 1 and 2). We appreciate again the careful check of reviewer #2 on our diffraction data to clarify the comparative stability of albite under different subduction conditions. We point out, however, that this new assignment would not affect our major findings of the hydration and dehydration breakdowns from albite along cold (wet) subduction conditions.

REVIEWERS' COMMENTS

Reviewer #1 (Remarks to the Author):

The revised manuscript has shown substantial modification made by the authors in response to the reviewers' questions. I noticed that they have added more calculations on the density and water content changes due to the phase change observed in the experiments. The density calculation will further allow geodynamicists to test the research's implications on mantle dynamics. They also proposed a new phase-change induced mechanism (hydration) that increases the seismicity at the depth of ~ 90 km depth. I am personally OK with the new terminology used here ("hydration faulting"), but I suggest that the authors to add a note for seismologists to test the hypothesis in the future somewhere in the summary (the mechanism hypothesizes that the increased seismicity is mainly from this hydration-related dynamic weakening process, and such phenomenon can perhaps be found in other subduction zones as well.)

Overall the revision improves the manuscript and makes it acceptable by Nature Communications.

Reviewer #2 (Remarks to the Author):

I am satisfied with the changes made by the authors on their original manuscript and by their detailed answers to my comments. Therefore I recommend this article for publication.

Reviewer #1 (Remarks to the Author)

Comments: *The revised manuscript has shown substantial modification made by the authors in response to the reviewers' questions. I noticed that they have added more calculations on the density and water content changes due to the phase change observed in the experiments. The density calculation will further allow geodynamicists to test the research's implications on mantle dynamics. They also proposed a new phase-change induced mechanism (hydration) that increases the seismicity at the depth of ~ 90 km depth. I am personally OK with the new terminology used here ("hydration faulting"), but I suggest that the authors to add a note for seismologists to test the hypothesis in the future somewhere in the summary (the mechanism hypothesizes that the increased seismicity is mainly from this hydration-related dynamic weakening process, and such phenomenon can perhaps be found in other subduction zones as well.)*

Overall the revision improves the manuscript and makes it acceptable by Nature Communications.

Reply: We appreciate the patience and continuing suggestions by reviewer #1. As suggested, we have added the statement in the discussion part as below (p7, line 186-189).

"We encourage future studies to test our hypothesized mechanism that the increased seismicity is mainly from this hydration-related dynamic weakening process, and such phenomenon can be found in other subduction zones."

Reviewer #2 (Remarks to the Author)

Comments: *I am satisfied with the changes made by the authors on their original manuscript and by their detailed answers to my comments. Therefore I recommend this article for publication.*

Reply: We appreciate very much the encouraging comments and support of Reviewer #2 for publication of our manuscript in Nature Communications.